

# The influence of a submarine canyon on the wind-driven downwelling circulation over the continental shelf

Pedro A. Figueroa [1], Gonzalo S. Saldías [1,2], and Susan E. Allen [3]

[1]Departamento de Física, Facultad de Ciencias, Universidad del Bío-Bío, Concepción, Chile
[2]Centro de Investigación Oceanográfica COPAS Coastal, Universidad de Concepción, Concepción, Chile
[3]Department of Earth, Ocean, and Atmospheric Sciences, University of British Columbia, Vancouver, Canada

**Correspondence:** Gonzalo S. Saldías (gsaldias@ubiobio.cl)

**Abstract.** The response of a coastal ocean model, simulating a typical Eastern Boundary system, to downwelling-favorable winds with and without the presence of a submarine canyon is studied. Three contrasting bathymetric configurations, considering shelves with different depth and slopes, are evaluated. Experiments without a submarine canyon represent the well-known downwelling circulation and cross-shore structure with a downwelling front and the development of frontal instabilities generating density anomalies in the bottom layer. The presence of the submarine canyon drives important changes in cross-shore flows, with opposing velocities on either side of the canyon. Onshore (offshore) and downward (upward) velocities develop in the upstream side of the canyon in the time-dependent and advective phases. Instabilities developed and are modified principally downstream of the canyon. Overall, the net impact of the canyon is to enhance offshore and downward transport. However, particle tracking experiments reveal that particles can become trapped inside the canyon in an anticyclonic circulation when the particles pass the canyon over the continental slope or when particles inside the canyon are affected by downwelling conditions. Overall, ∼20-23% (∼15-18%) of particles released directly upstream (in the canyon) at mid-depths become trapped inside the canyon until the end of the simulations (15 days).

## 1 Introduction

Submarine canyons are common features along continental shelves, acting as pathways for water exchange between the continental slope and the coastal ocean by means of upwelling and downwelling dynamics, which facilitates the cross-shore exchanges of heat, nutrients and organic matter. These cross-shore exchanges are tied to the circulation induced by the canyon as it breaks the along-isobath flow, generating ageostrophic circulation and enhanced cross-shore transports (Allen and Durrieu de Madron, 2009; Saldías and Allen, 2020). The effects of submarine canyons on circulation have been studied with in-situ observations (e.g., Hickey, 1997; Allen et al., 2001; Flexas et al., 2008; Sobarzo et al., 2016; Brun et al., 2023), modeling approaches (e.g., Klinck, 1996; She and Klinck, 2000; Jordi et al., 2005; Spurgin and Allen, 2014; Ahumada-Sempoal et al., 2015; Saldías and Allen, 2020), and theoretical scaling analysis (Allen and Hickey, 2010; Howatt and Allen, 2013). Modeling efforts have shown that the circulation around a submarine canyon depends primarily on the direction of the incident flow, generating different patterns of circulation in the presence of right or left-bounded flows. For upwelling favorable flow (left/right-bounded in the Northern/Southern Hemisphere), the canyon enhances the upwelling, driving a pool of dense water on the shelf downstream of





the canyon (Saldías and Allen, 2020). In contrast, for downwelling favorable flow (right/left-bounded in the Northern/Southern Hemisphere) promote an anti-symmetrical circulation, with onshore and downward velocities along the upstream side and offshore/upward velocities in the downstream side of the canyon (Klinck, 1996; Spurgin and Allen, 2014). The patterns of circulations are further modified by the stratification in the canyon, with contrasting patterns related to the Burger Number (Spurgin and Allen, 2014), and the orientation of the canyon relative to the incident flow (Ahumada-Sempoal et al., 2015;

Wang et al., 2022).

Regardless the importance of submarine canyons as conducts of cross-shore transport between the deep and coastal oceans and its related ecological characteristics, these sites are still poorly sampled and studied. Observational studies are difficult due to the complex topography and the common presence of fisheries leading to the loss of instruments, which have raised the importance of theoretical and modeling approaches. These modeling studies have given insights on the circulation around

these sites in idealized (e.g., Klinck, 1996; She and Klinck, 2000; Kämpf, 2006; Spurgin and Allen, 2014; Zhang and Lentz, 2017; Saldías and Allen, 2020) and realistic conditions (Jordi et al., 2005; Connolly and Hickey, 2014; Ahumada-Sempoal et al., 2015; Wang et al., 2022). However, these studies have been primarily focused in upwelling conditions around submarine canyons (e.g Klinck, 1996; Kämpf, 2006; Howatt and Allen, 2013; Connolly and Hickey, 2014; Saldías and Allen, 2020), and downwelling conditions have been significantly less studied. The few modeling studies that have addressed the downwelling

response have presented two types of circulation, controlled by the Burger Number at the canyon (Spurgin and Allen, 2014). Furthermore, the idealized studies have used a body-force forcing condition (Klinck, 1996; She and Klinck, 2000), or a combination of body-force and local wind forcing (Spurgin and Allen, 2014), while the realistic setups have resolved downwelling conditions by means of downwelling currents (Jordi et al., 2005; Ahumada-Sempoal et al., 2015). In the case of downwelling wind-driven experiments, the influence of a shelf valley, shallower than a submarine canyon, has been studied (Zhang and

Lentz, 2017), as well as the impact of a submarine canyon using only surface wind-forcing (Spurgin and Allen, 2014). However, the response of the flow field to deep and steep submarine canyons forced only by local downwelling winds and with different continental slopes, typical of Eastern Boundary Systems, has not been addressed yet in detail.

The upwelling and downwelling effects of a submarine canyon in the local circulation are different, with a stronger response for upwelling-favorable conditions than for its downwelling counterpart (Klinck, 1996; Zhang and Lentz, 2017). However, high

productivity and biological hotspots have been observed and predicted in downwelling submarine canyons (Skliris and Djenidi, 2006; Flexas et al., 2008; Mordy et al., 2019) even when the net effect of submarine canyons in downwelling conditions is to enhanced the downwelling response (Klinck, 1996; She and Klinck, 2000). This biological characteristics have been hypothesized to be linked to local onshore and positive vertical velocities in downwelling canyons, as well as the effects of the cyclonic and anticyclonic circulations that could induce the trapping of particles (Skliris and Djenidi, 2006; Spurgin and Allen,

2014). In a realistic simulation in Blanes Canyon in the NW Mediterranean Sea, Ahumada-Sempoal et al. (2015) found that particles tend to aggregate in the head of the canyon by the presence of a seasonal varying downwelling current. Furthermore, submarine canyons could be connected by means of particle transport in a regional context in downwelling conditions, which could be important for pelagic larval stages (Clavel-Henry et al., 2019). However, for wind-driven downwelling events, as those developed in Eastern Boundary Systems by synoptic storm events, this trapping of particle has not been studied. This





is an important feature considering that submarine canyons have also been associated with biological hotspots and enhanced productivity compared to the adjacent slopes (e.g. Allen et al., 2001; Skliris and Djenidi, 2006; De Leo et al., 2010; Ceramicola et al., 2015; Santora et al., 2018).

The present study aims to further understand the induced effects of a submarine canyon in the context of wind-driven downwelling forcing, and how these effects are observed in the local circulation, cross-shore and vertical transports, and the

potential of trapping and onshore transport of particles in the coastal ocean. A high-resolution numerical model forced with constant downwelling-favorable winds and variable bathymetric configurations is used with and without a submarine canyon. These simulations are also analyzed in a Lagrangian particle tracking experiment to study the effects of the canyon on particles along the shelf. Section 2 presents the details of the model configurations and particle experiments, section 3 describes the main results, section 4 gives a detailed discussion about our findings and the final conclusions are given in section 5.

## 70  2  Methodology

### 2.0.1  Model Configuration and Experiments

We used the Regional Ocean Modeling System (ROMS) (Shchepetkin and McWilliams, 2005), a primitive equation model that uses finite differences to resolve the hydrostatic primitive equations. Vertical differencing is achieved with terrain following (sigma) coordinates (Song and Haidvogel, 1994). The model is run with a third-order upstream horizontal and a fourth-order

centered vertical advection scheme for momentum and tracers to avoid spurious vertical velocities at the canyon rim due to stratified flow over steep topography using sigma coordinates (Dawe and Allen, 2010). The horizontal pressure gradient is treated with a spline density Jacobian (Shchepetkin and McWilliams, 2003). Vertical mixing was computed using Mellor–Yamada level 2.5 closure scheme (Mellor and Yamada, 1982) and the background vertical viscosity and diffusivity were set to $1 \times 10^{-5}$ and $5 \times 10^{-6} m^2 s^{-1}$, respectively. Bottom stress is calculated with a quadratic drag law using a bottom roughness of $2 \times 10^{-2} m$.

The experiments consisted in an idealized coastal ocean, typical of an eastern boundary system, with dimensions of 155 km in the cross-shore direction and 600 km in the alongshore direction, with higher resolution near the coast that diminishes offshore. Grid spacing is 0.5 km in both cross-shore and alongshore directions in the region around the canyon (Fig. 1b). From 50 km offshore the grid resolution decreases towards the western boundary with a maximum grid spacing of 10 km. Three types of bathymetric configurations were used by changing the shelf slope. The maximum depth was 500 m offshore in

all configurations. Deep shelf experiments were based on the bathymetry used by Klinck (1996) and derived works, where a canyon is defined as:

$$H(x,y) = H_m - \frac{H_s}{2} \left[ 1 - \tanh \frac{-x - x_o(y)}{a} \right] \tag{1}$$

where $H_m$ is the maximum depth of the domain (500 m), $H_s$ is the depth change from the continental shelf to the open ocean (400 m), $a$ is the transition scale defining the slope of the cross-shelf profile (5 km), and $x_o(y)$ is the location of the shelf break,

defined as:



$$x_o(y) = x_n + x_b \left[ 1 - \exp \frac{-(y^2 - y_o^2)}{2b^2} \right] \tag{2}$$

where $x_n$ is the nominal distance of the head of the canyon from the coastal wall (12 km), $x_b$ is the distance added to $x_n$ to reach the shelf break (10 km), $y_o$ is the location of the center of the canyon (at y = 0 km), and $b$ is the width scale of the canyon (2.5 km). This configuration produces a canyon 10 km wide in its mouth, 20 km long from its mouth to the head, and with sidewall steepness of $\sim$0.065. Intermediate and Shallow experiments were run with more realistic sloping shelves by changing some key parameters in (1) and (2) as in Saldías and Allen (2020). The experiments were configured with 30 s-coordinate levels, using an increased resolution near the surface and bottom to resolve the boundary layers. The domain has three open boundaries (north, south, and offshore), where implicit gravity wave radiation conditions (Chapman, 1985) and the Flather radiation scheme (Flather, 1976) were applied to the surface elevation and depth-averaged horizontal velocities, respectively. Orlanski radiation conditions (Orlanski, 1976) were applied to the baroclinic velocities, temperature, and salinity on the western boundary. A local two-dimensional model of the northern and southern boundaries was run to extract the local fields for those boundary conditions (e.g. Gan and Allen, 2005; Castelao and Barth, 2006; Saldías and Allen, 2020).

The six experiments (3 bathymetries, canyon and no-canyon cases) were forced only with a horizontally uniform surface wind stress that is ramped up from 0 to 0.03 $Nm^{-2}$ in 5 days (from day 10 to 15), and then maintained constant for the following 10 days of the simulations. The first 10 days of the model were run free to let transients decay. Initial conditions in temperature and salinity are the same as those used by Saldías and Allen (2020) based on the average stratification conditions off Oregon (see Fig. 1c).

Dimensionless numbers are calculated for each experiment and summarized together with bathymetric constant and wind forcing in Table (1). The vertical aspect ratio $H_s/L$ is calculated defining $H_s$ as the depth of the canyon from the shelf to the bottom of the canyon and $L$ is the length of the canyon, defined using the shallowest isobath related to the canyon rim. A description of the incoming velocity interacting with the canyon is given by the Rossby Number ($R_o$):

$$R_o = \frac{U}{fL} \tag{3}$$

where $U$ is the incoming velocity interacting with the canyon or shelf (i.e, upstream of the canyon location), calculated by averaging the meridional velocity over the continental shelf, $f$ is the Coriolis Parameter and L is the length of the canyon or shelf width. Another version of the Rossby Number but using the radius of curvature $R$ rather than the length of the canyon is also calculated ($R_{oc}$) following Howatt and Allen (2013). In addition, a description of the stratification conditions is given by the Burger Number $Bu$:

$$Bu = \frac{NH_{sb}}{fL} \tag{4}$$



**Table 1.** List of experiments with and without a submarine canyon under contrasting bathymetric configurations. $H_s/L$ is the vertical aspect ratio (depth of the shelf break over the length of the canyon). $R_o$ and $R_{oc}$ are the Rossby Number using the length of the canyon/shelf and the radius of curvature, respectively.

| Exp | Exp1 | Exp2 | Exp3 | Exp4 | Exp5 | Exp6 |
|---|---|---|---|---|---|---|
| Canyon | No | Yes | No | Yes | No | Yes |
| Shelf | Deep Shelf | Deep Shelf | Intermediate Shelf | Intermediate Shelf | Shallow Shelf | Shallow Shelf |
| $\|\tau_y\|(Nm^{-2})$ | 0.03 | 0.03 | 0.03 | 0.03 | 0.03 | 0.03 |
| $H_m(m)$ | 500 | 500 | 500 | 500 | 500 | 500 |
| $H_s(m)$ | 400 | 400 | 446 | 446 | 486.5 | 486.5 |
| $x_n(x10^3 m)$ | 12 | 12 | 12 | 12 | 12 | 12 |
| $x_b(x10^3 m)$ | 10 | 10 | 10 | 10 | 10 | 10 |
| $a(x10^3 m)$ | 5 | 5 | 10 | 10 | 10 | 10 |
| $b(x10^3 m)$ | 2.5 | 2.5 | 2.5 | 2.5 | 2.5 | 2.5 |
| $H_s/L$ | - | 0.01 | - | 0.009 | - | 0.005 |
| $R_o$ | 0.143 | 0.187 | 0.178 | 0.196 | 0.198 | 0.165 |
| $R_{oc}$ | – | 0.4 | – | 0.43 | – | 0.37 |
| $Bu$ | – | 1.323 | – | 0.774 | – | 0.222 |
| $S$ | 0 | 0 | 0.126 | 0.143 | 0 | 0.07 |

where $N$ is the mean buoyancy frequency over the upstream continental shelf, $H_{sb}$ is the depth of the shelf, and $f$ and $L$ are
the same as for the Rossby Number. A complementary version of $Bu$, called the topographic Burger number is calculated as
follows:

$$S = s\frac{N}{f} \tag{5}$$

where $s$ is the continental shelf slope, calculated by taking the slope between the depth at the beginning of the shelf break and
the coast.


## 2.1   Particle Experiments

Horizontal and vertical velocity fields from the ROMS experiments were used as inputs for offline particle tracking experiments
using the Parcels v2 code (Delandmeter and Van Sebille, 2019). These Lagrangian experiments were run to assess the effects
of canyon-driven circulation on the transport of particles in the canyon and along the continental shelf. Parcels (Probably A
Really Computationally Efficient Lagrangian Simulator) is a Python based particle tracking module that allows the release of



**Figure 1.** (a) Model domain bathymetry (colors; m) with the location of the submarine canyon enclosed in a red box. (b) Zoom in to the area of the submarine canyon enclosed in (a), with the 100, 125, 150, 200, 250, 300, 350, 400 and 450 m isobaths shown in red contours. Yellow and green lines indicate the vertical planes where particles were released for particle tracking experiments. (c) Initial conditions in density (green line) and stratification (buoyancy frequency, red line) imposed for the whole domain shown in (a).





passive and active particles using outputs of ocean circulation models. We defined three areas of release based on the main characteristics of the canyon-induced circulation: 1) 3 zonal planes upstream of the canyon (y = -20 km, y = -30 km & y = -40 km) reaching from near the coast to the rim of each canyon configuration (-30 km < x < -5 km) and from the surface to the maximum depth of the continental shelf, 2) meridional planes at x = -30 km, x = -20 km, x = -10 km and x = -5 km, and

extending in the meridional direction from y = -20 km to y = 20 km and from the surface down to the bottom, with a maximum depth of 500 m in x = -30 km., 3) Lastly, particles were released inside the canyon at different horizontal planes, starting at the rim depth and reaching down to the bottom of the canyon for each bathymetric configuration. These planes extent from x = -5 km to x = -20 km and y = -10 km to y = 10 km in the horizontal. The particles were configured using a fourth-order Runge-Kutta advection scheme that allows vertical displacement. The particles were defined as non-buoyant elements and

without diffusivity. They were released at day 10 of each simulation and were tracked until the end of the simulations (day 25).

## 3 Results

### 3.1 Downwelling patterns: Canyon vs No-Canyon experiments

Variations in downwelling patterns are influenced by slope configuration and the presence of a submarine canyon (Fig. 2). In deep shelf experiments lacking a canyon, downward depth-averaged velocities occur near the coast, attributed to water descent

from downwelling winds (Fig. 2a) and horizontal velocities are predominantly northward. In the case of intermediate and shallow shelf experiments (Fig. 2b,c), instabilities emerge farther offshore, with intensified vertical velocities along the coast in the location of the downwelling front.

The presence of a submarine canyon induces notable changes in its vicinity (Fig. 2d,e,f). In the deep shelf experiments, downward velocities occur upstream of the canyon, while upward velocities prevail downstream. These vertical velocities ex-

tend up to 15 km offshore in a cross-shore band and exhibit a near-symmetrical distribution. A slight cyclonic deflection in horizontal velocities is observed over the canyon. Frontal instabilities persist in the intermediate and shallow shelf experiments, with intensified vertical velocities near the canyon (Fig. 2e,f). In the intermediate shelf experiment, interactions of these instabilities with the canyon alter the canyon-induced dipole of vertical velocities seen in the deep shelf experiment, generating pronounced vertical motions from the coast up to 8 km offshore (Fig. 2e). In the shallow shelf experiment, a stronger dipole

of vertical velocities, compared to the other experiments, is evident from the coast to approximately 10 km offshore, with depth-averaged vertical velocities stronger than in the deep shelf experiment. While instabilities are modified downstream, the extent of alteration is less pronounced than in the intermediate shelf experiment.

The cross-shore structure of downwelling circulation exhibits variations with changes in bathymetry (Fig. 3). In the deep shelf experiments (Fig. 3, left column), surface and bottom Ekman layers are discernible in cross-shore velocities, featuring

onshore transport at the surface and offshore transport at the bottom on the shelf. Isopycnals tilt downwards, with the formation of a downwelling front near the coast by day 25. Northward flow prevails across the domain, characterized by two primary jets in the cross-shore direction. One lies over the shelf-break at around 100 m below the surface, whereas a second, stronger jet, develops next to the coast, and extends through most of the water column. As bathymetry shallows in the intermediate





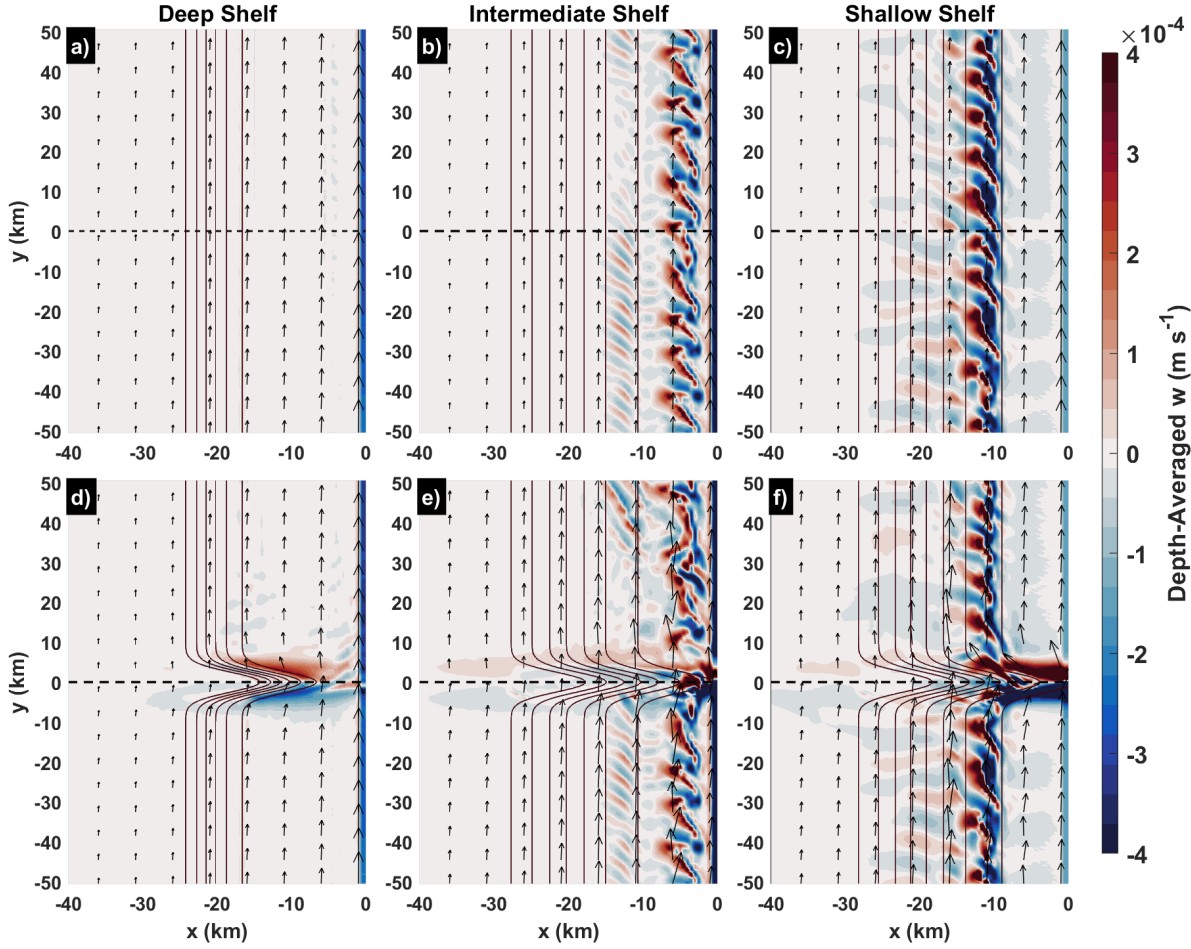

**Figure 2.** Depth-Averaged vertical velocities (color) and depth-averaged horizontal currents (arrows) for (a, c) Deep, (b, e) Intermediate, and (c, f) Shallow shelf configurations with (lower panels) without (upper panels) a submarine canyon.

and shallow experiments, instabilities modify the cross-shore downwelling structure and the position of the northward jet (Fig.

3, central and right columns). The downwelling front is located around x = -5 km and x = -10 km in the intermediate and shallow shelf experiments, respectively, coinciding with the bending of isopycnals towards the bottom. These positions align with regions of strong vertical velocities (Fig. 2), indicating that instabilities do not form in deep shelf experiments by day 25 due to incomplete development of the downwelling front. These instabilities alter the Ekman layers, disrupting Ekman transport by enhancing or reducing velocities within these layers. Regarding northward flow, relative zones of strong velocities

are observed around the downwelling front in the intermediate shelf case, at x = -5 and x = -10 km, whereas the downwelling jet emerges just above the downwelling front at x = -10 km in shallow shelf experiments.

The influence of a submarine canyon induces notable alterations in cross-shore velocities (Fig. 4). Upstream, three layers of cross-shore velocities emerge in the deep shelf experiment (Fig. 4m): the surface onshore Ekman layer, the bottom offshore





**Figure 3.** Cross-shore sections of velocity (color) and density (gray lines) fields at 15 km (a-f), 0 km (g-l) and -15 km (m-r) for the no-canyon simulations at day 25.




Ekman layer, and an intermediate layer with weaker onshore velocities. While this intermediate layer is consistent across all configurations, its characteristics are modified by instabilities in the intermediate and shallow shelves. Alongshore velocities remains almost unchanged upstream of the canyon compared to the no-canyon counterparts in all bathymetric configurations. Along the canyon axis (Y = 0 km), offshore and onshore velocities occur within the canyon. In the intermediate and shallow cases, instabilities extend into the canyon, inducing cross-shore exchanges. Isopycnals evolve from a tilted orientation within the canyon to nearly horizontal farther offshore. Alongshore velocities show a weak counter-current within the canyon in the intermediate and shallow shelf experiments with opposing velocities outside the canyon aligned with the change in isopycnal orientation. Downstream (Fig. 4a-f), a layer of offshore velocities extends from the base of the surface Ekman layer to the bottom. While instabilities modify this layer near the coast in the intermediate and shallow cases, it extends well offshore in all experiments, suggesting that the canyon presence influences the circulation farther offshore. Notably, there are no significant changes in the cross-shore position of the downwelling jet after it passes the canyon (Fig. 3d,e,f versus Fig. 4d,e,f).

The presence of a canyon is linked to an increase in the Rossby Number in the deep and intermediate experiments, but not in the shallow experiment, as short length scales are already present (Table 1). In general, flows in all six experiments are geostrophic. In terms of the stratification, the presence of a canyon does not make a large difference. However, values tend to diminish as the water shallows over the continental shelf, with $Bu = 0$ and $S = 0$ in the shallow shelf experiments, related to the density homogenization in the area selected to compute the numbers caused by the faster evolution and displacement of isopycnals in this configuration compared to deep and intermediate shelf experiments.

Alongshore sections at two distinct locations (x = -6 km and x = -13 km) elucidate the canyon's impact on the vertical structure of the cross-shore velocities (Fig. 5). These locations were chosen based on the cores of higher variability observed in Fig. 2. In the absence of a canyon, the surface and bottom Ekman layers are clearly discernible in the deep shelf experiment (Fig. 5a,g). However, the presence of a canyon induces a dipole of opposing cross-shore velocities over the canyon walls (Fig. 5d,j), disrupting the bottom Ekman layer over the canyon walls. Depending on location, these velocities may extend through the entire water column. At x = -13 km, cross-shore velocities are confined to the shelf near the canyon walls, without extending to the canyon bottom. Two weak counter-currents develop inside the canyon, generating contrasting circulation patterns. Offshore and onshore velocities induced by the canyon extend to the surface at this location, interacting with the surface Ekman layer. At x = -6 km, velocities extend throughout the entire canyon, with offshore velocities on the downstream side penetrating the base of the surface Ekman layer (Fig. 5j). These offshore velocities extend approximately 25 km north of the canyon and are associated with a tilted isopycnal on the downstream side.

Cross-shore velocities primarily relate to instabilities in the intermediate experiments without a canyon (Fig. 5h). In the presence of a canyon, an anticyclonic circulation forms inside the canyon at x = -13 km, beneath a weak cyclonic dipole interacting with instabilities. Instabilities reappear approximately 30 km north of the canyon. At x = -6 km, instabilities dominate cross-shore velocities even in the presence of the canyon. Notably, there is a distinct modification of their structure as they pass over the canyon, with the patterns being different on the downstream compared to the upstream side. In the shallow shelf experiments, instabilities dominate the cross-shore velocity structure at x = -13 km. The presence of a canyon induces a cyclonic (anticyclonic) circulation over (inside) the canyon (Fig. 5f). Downstream, instabilities are minimally affected by the



**Figure 4.** Cross-shore sections of velocity (color) and density (gray lines) fields at 15 km (a-f), 0 km (g-l) and -15 km (m-r) for the canyon simulations at day 25





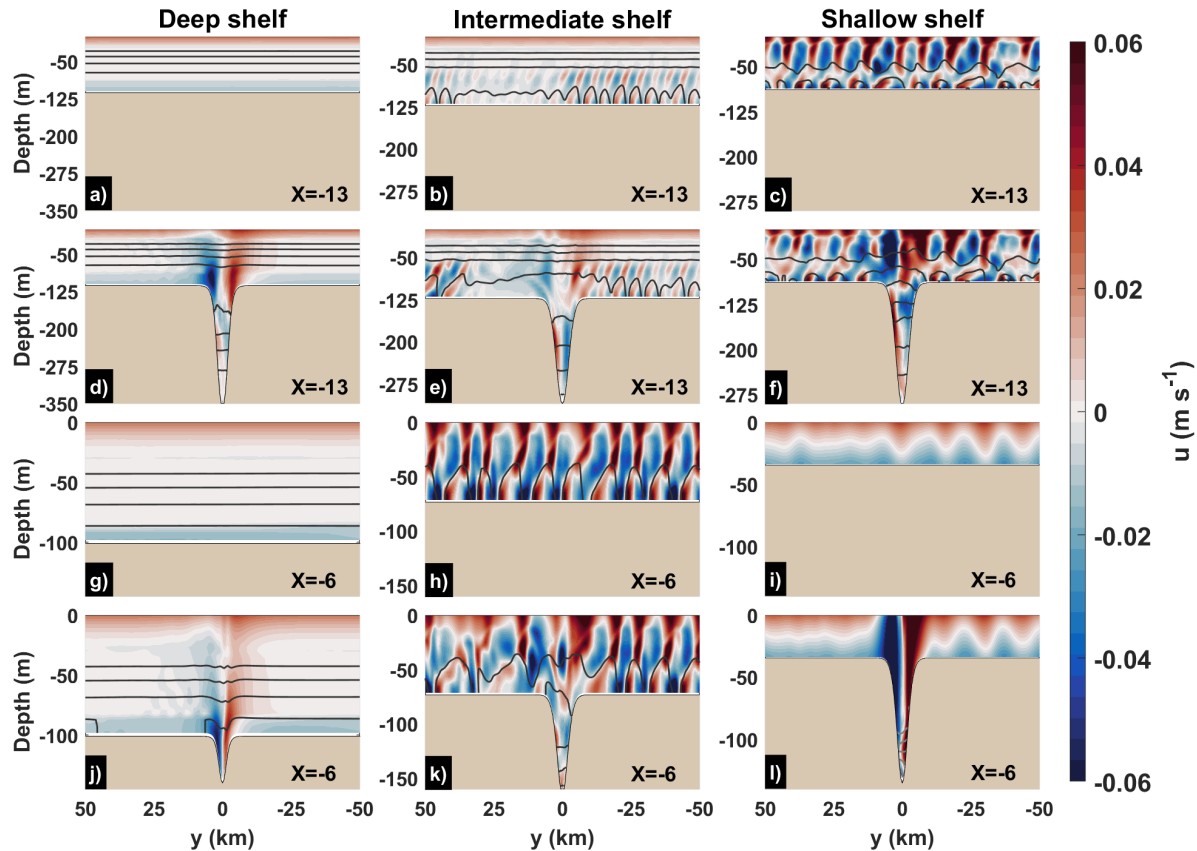

**Figure 5.** Alongshore sections of velocity (color) and density (gray lines) fields at x = -13 km (a-f) and x = -6 km (g-l) comparing the no-canyon (a-c;g-i) and canyon (d-f;j-l) experiments at day 25.

canyon. However, the nearshore cross-shore circulation (x = -6 km) mirrors that of the deep shelf experiments, as instabilities
are located farther offshore. The presence of a canyon disrupts bottom and surface Ekman layers, generating strong onshore (offshore) velocities upstream (downstream) of the canyon, affecting the entire water column (Fig. 5l).

The effects of the canyon are evident in the density variations observed between canyon and no-canyon runs at the end of the simulations. In the deep shelf experiments, a layer of lighter water is observed over the downstream shelf in the presence of the canyon (Fig. 6a), confined to the lower 20 meters and followed by a layer of higher density water around 20-80 meters
from the surface, much clearer in x = -13 km (Fig. 6d,g). The difference in density near the coast is more pronounced, with faster advection of light surface water to the bottom in the presence of the canyon, resulting in lighter water compared to the no-canyon experiment at a given time. A column of negative anomalies also appears over the canyon axis, related to the extended downward tilt of isopycnals in the canyon cases.

In the intermediate and shallow shelf experiments, the density differences clearly indicate modifications to the instabilities
by the canyon. Upstream, there are no significant differences between the canyon and no canyon experiments. However, after





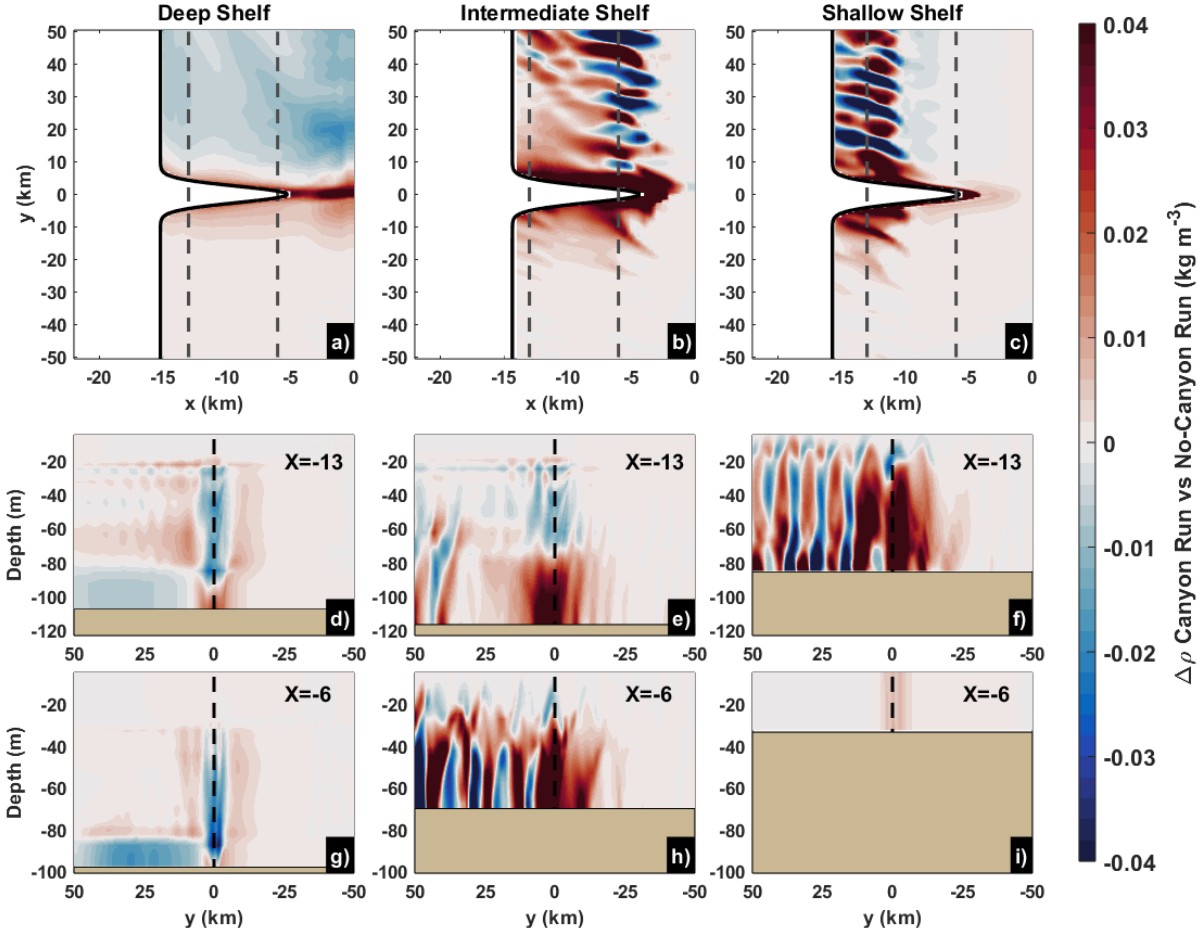

**Figure 6.** Density differences between the Canyon Runs (Exp 8,10,12) and No-Canyon Runs (Exp 7,9,11). a-c) Differences at the closest level to the bottom for each configuration. d-i) Alongshore section at the gray lines showed in (a-c) at x = -13 km and x = -6 km

passing the canyon, positive and negative density anomalies are observed in both experiments, centered at x = -6 km in the intermediate configuration and at x = -13 km in the shallow configuration. These density differences align with the positions of the instabilities shown in Fig. 2, and only appear downstream as upstream the instabilities have not yet interacted with the canyon. As these instabilities pass over the canyon, their wavelengths are altered, leading to local density anomalies compared to the distribution in the no-canyon scenarios. In the intermediate shelf experiments, these instabilities affect up to 20 meters below the surface near the coast, while in the shallow configuration, differences in density extend up to the surface at x = -13 km, but are minimal near the coast.





## 3.2 Cross-shore and vertical transports

The spatial extent and magnitude of the cross-shore transport are significantly altered by the presence of a submarine canyon
(Fig. 7). In the absence of a canyon, the primary cross-shore velocities consist of the Ekman layers during the advective phase,
along with velocities induced by instabilities in the intermediate and shallow shelf experiments. Consequently, the deep shelf
experiment exhibits an increasing onshore transport over the course of the simulation due to an imbalance between the surface
and bottom Ekman layers (Fig. 7e), resulting in a high cumulative onshore transport by the end of the simulation. In contrast,
the intermediate and shallow shelf experiments evolve more rapidly, achieving a balance between the Ekman layers and having
low cross-shore transports throughout the simulation.

In the case of the shallow shelf at x = -13 km, the presence of instabilities is evident in the variable cross-shore transport
along the coast, resulting in fluctuating magnitudes, although with minimal impact on cumulative transport by the end of the
period (Fig. 7b). The presence of the canyon induces a reduction in onshore transport in all three simulations (Fig. 7c,d). The
temporal evolution shows an increase in offshore transport in all experiments in response to velocities generated around the
canyon, transitioning to the advective phase where cross-shore transport becomes positive in the deep-shelf experiment and
close to zero in the intermediate shelf experiment. In contrast, the net transport remains negative in the shallow shelf experiment,
reflecting the influence of instabilities in the canyon experiment, but without reaching positive values and exhibiting a trend
towards net offshore transport over the period. This trend is further highlighted in the cumulative transport in the canyon run,
where the intermediate and shallow shelves consistently show a consistent offshore transport, while the deep shelf experiment
exhibits a net onshore transport from about day 21 onwards.

The spatial distribution of cross-shore transport underscores the significance of canyon-induced velocities (Fig. 7e,f). In the
absence of a canyon, cross-shore transport remains relatively constant with minimal variations alongshore, attributed to the
balance between Ekman layers. In the shallow case, where instabilities are fully developed at x = -13 km, high alongshore vari-
ability is observed. The presence of a canyon results in strong onshore transport upstream and offshore transport downstream
of the canyon, with the greatest magnitudes observed in the deep shelf experiment.

The vertical transport induced by the canyon at the shelf level (Fig. 8) exhibits an antisymmetrical response on the down-
stream and upstream sides of the canyon, consistent with the depth-averaged vertical velocities. By analyzing the net vertical
transport into upstream (from y = -6 km to y = 0 km) and downstream (from y = 6 km to y = 0 km) components, it is evident
that both sides of the canyon induce vertical transport of nearly equal magnitudes. Following day 20, fluctuations in vertical
transport are observed in the intermediate and shallow shelf experiments, linked to the development and passage of instabilities
near the canyon (Fig. 8c,e).

In all three simulations, the initial five days are characterized by a negative net transport, corresponding to negative velocities
on both sides of the canyon from day 10 to 13. Subsequently, a sign change occurs in the downstream side, while the upstream
side maintains negative values, leading to a quasi-balance between upwelling and downwelling. The vertical transport stabilizes
and remains relatively constant until the instabilities develop in the intermediate and shallow experiments. Notably, in the
shallow shelf case, vertical transport even approaches zero between days 16 and 20, indicating a balance between upwelling





**Figure 7.** Cross-shore transport for all experiments at the plane farther from the coast shown in Fig. 6-7 (x = -13 km). a-b) Cross-shore and Cummulative transport for the No-canyon experiments. c-d) Same as before, but for the canyon experiments. e-f) Along-shore distribution of the Cross-shore transport for the no-canyon and canyon simulations. Note that the cross-shore transport is two order of magnitudes greater in the canyon simulations (f) than its no-canyon counterpart (e), and different scales are shown for a better description of the patterns in each case



and downwelling (Fig. 8e). However, the overall effect of the submarine canyon results in a net downwelling throughout the simulation. The deep shelf experiment exhibits greater vertical transports compared to the intermediate shelf experiment (Fig. 8b,d).

## 3.3 Fate of particles under downwelling conditions

The velocity perturbations induced by the canyon result in significant cross-shore transport and vertical velocities around the canyon, contrasting with the experiments lacking a submarine canyon. An examination of particle dispersion along the shelf in the presence of a canyon reveals that particles can travel up to 15 km in the cross-shore direction along the canyon (Fig. 9). In contrast, without a canyon, particles' cross-shore displacement reaches a maximum of 2 km outside the surface and bottom Ekman Layer, a trend consistent across all three bathymetric configurations.

Particles with a the canyon are more widely dispersed compared to configurations without a canyon, although their along-shore distance remains relatively unchanged. For particles initially positioned upstream of the canyon (20 km upstream), the presence of the canyon induces horizontal deflection in their trajectories throughout the water column (Fig. 9). This effect manifests as a cyclonic turn over the canyon following the isobaths, with particles returning to their original cross-shore positions after passing over the canyon, particularly for particles above the shelf. Conversely, particles at shelf-break depth and below tend to follow the isobaths and can become trapped within the canyon, exhibiting anticyclonic circulation patterns extending up to 15 km in length (Fig. 9a-c). The various slope configurations determine the depth at which particles may become trapped inside the canyon, as well as the quantity of particles entrapped. Steeper slopes tend to trap more particles than relatively flat shelves, with particles potentially remaining confined within the canyon for up to 15 days. Notably, particles released upstream at y = -30 km and y = -40 km did not show significant trapping within the canyon at any depth (not shown).

Particles released outside the canyon rim or parallel over the shelf (yellow planes in 1) were influenced by the presence of the canyon (Fig. 9d-f), resulting in advection inside the canyon on the upstream side and subsequent advection outside the canyon on the downstream side. The particles were then carried northward by the geostrophic current, leading to dispersion compared to their initial positions. This effect is more pronounced with increasing depth and shelf slope, with particles following isobaths as they enter the canyon and traveling cross-shore up to 4 km, although they were not observed to become entrapped in any simulation.

Particles released at shelf-break depth, or within the canyon and parallel to the rim, either became trapped within the canyon with anticyclonic circulation or continued following isobaths before exiting at the canyon mouth, being advected northward by the current. Trapped particles within the canyon traveled up to 7 km toward the coast. Overall, the vertical displacements were generally negative, with the vertical velocities induced by the submarine canyon not significantly affecting the vertical particle movements. Cross-shore velocities were identified as the primary pathway for trapping particles within the canyon. For particles leaving the canyon on the downstream side, they tended to aggregate farther offshore from their initial release positions when starting at shelf-break depth or below.

Overall, around a 20% of particles become trapped up to the end of the experiments when they were released upstream (Fig. 9a-c) at y = -20 km at mid-depths (115-185 m, 100-170 m, 75-145 m for the deep/intermediate/shallow shelf experiments).







**Figure 8.** Vertical transport for the downstream and upstream areas in the three simulations with a submarine canyon, calculated at the depth of the shelf in the zonal extension between the coast and x = -20 km. (left panels) Vertical and (right panels) Cumulative transport at (a,b) z = -100 m for the Deep shelf canyon, (c-d) at z = -60 m for the Intermediate shelf canyon, and (e-f) at z = -20 km for the Shallow shelf canyon. Every panel presents different scales in order to highlight the local vertical transports and their variability.





**Figure 9.** Examples of particle trajectories around a canyon for the three shelf configurations (Deep Shelf – DS; Intermediate Shelf – IS; Shallow Shelf – SS). The color indicates the time since its release, whereas the white points denote their initial position. (Upper panels) Particle trajectories released in a zonal plane at y = -20 km. (Bottom panels) Particles trajectories released in a meridional planes at x = -20 km and -20 km ≤ y ≤ 20 km.





The percentage of particles appeared to increased with increasing slopes and decreasing shelf depths. Outside this range, the percentage of particles trapped inside the canyon at the end of the simulation were lower, thus indicating a reduction of particles trapped outside these boundaries. In the case of the particles parallel to the coast at x = -20km (Fig. 9d-f), between 15.5-18.5% of particles released at 210-260 m, 200-250 m, 190-240 m for the deep/intermediate/shallow shelf experiments respectively, were trapped inside the canyon, with no clear pattern related to bathymetric changes.

## 4 Discussion

### 4.1 Downwelling circulation and canyon effects

Wind-driven downwelling is characterized by onshore Ekman transport, leading to an offshore pressure gradient force and poleward current, along with offshore transport in the bottom Ekman layer in eastern boundary systems. Previous studies have demonstrated that the presence of a submarine canyon under downwelling conditions can amplify downwelling and alter local circulation (Klinck, 1996; She and Klinck, 2000; Skliris et al., 2001, 2002), resulting in onshore (offshore) and downward (upward) velocities on the upstream (downstream) side in both idealized (Klinck, 1996; Spurgin and Allen, 2014; Zhang and Lentz, 2017) and realistic simulations (Jordi et al., 2005; Ahumada-Sempoal et al., 2015). Our findings indicate a consistent presence of positive vertical velocities on the downstream side of the canyon, persisting almost continuously but subject to variations induced by passing instabilities over the canyon.

Moreover, notable onshore velocities are observed along the upstream side of the canyon, particularly around the rim, aligning with prior research. These canyon-induced velocities can induce local upwelling even in regions experiencing overall downwelling conditions, facilitated by onshore up-slope transport. However, the overall vertical transport around the canyon remains predominantly downward, with the upstream downward component slightly exceeding the downstream upward component. Additionally, the net cross-shore transports demonstrate a consistent offshore trend during time-dependent and advective phases in all three experiments. This suggests that the collective impact of the canyon serves to intensify downward and offshore velocities, reinforcing the downwelling process.

Previous studies examining downwelling conditions in idealized settings predominantly utilized different approaches to force the circulation, including body-force forcing to simulate remote wind forcing scenarios (Klinck, 1996), local surface wind forcing (Spurgin and Allen, 2014), as well as combinations of wind forcing and body-force forcing (She and Klinck, 2000; Spurgin and Allen, 2014). These studies aimed to simplifying the analysis while omitting the surface and bottom Ekman layers. In contrast, our study introduces a novel contribution by focusing on a pure wind-forced circulation approach. We observe local modifications in surface onshore and bottom offshore transport, influenced by the presence of a submarine canyon. Specifically, the onshore velocities observed on the upstream side of the canyon disrupt the bottom Ekman layer, a pattern particularly evident in both the Deep and Shallow experiments. These cross-shore velocities are visible during early stages of the experiments due to canyon-induced effects (as seen on the deep shelf experiment for day 25) and later during more developed conditions, due to both effects of instabilities interacting with the submarine canyon and the canyon effect (as seen on the intermediate and shallow experiments). Similarly, offshore velocities on the downstream side can extend to





the surface Ekman layer near the coast. Beyond the canyon area, the canyon's presence generates onshore velocities beneath
the Ekman layer in the upstream region, and offshore velocities extending from the bottom Ekman layer up to the base of the
surface Ekman layer in the downstream region across all three configurations. The enhancement of offshore velocities on the
downstream side may explain the observation of less dense water in that area. Furthermore, the accelerated offshore transport
induced by the canyon causes the advection of surface water to occur more rapidly compared to scenarios without a canyon.

Three types of continental shelf slopes were examined, with the final five days of our experiments facilitating the develop-
ment of instabilities. Previous studies have primarily focused on the impact of changes in continental shelf slope on submarine
canyons under upwelling conditions, utilizing scaling analyses (Howatt and Allen, 2013) and idealized numerical simulations
(Saldías and Allen, 2020). Studies addressing frontal instabilities over a submarine canyon have similarly concentrated on up-
welling conditions (Saldías and Allen, 2020). These investigations have demonstrated that steeper slopes and increased stratifi-
cation lead to a shallower depth of upwelled water and reduced upwelling flux, with the velocity of incoming flows exhibiting
the opposite trend (Howatt and Allen, 2013). Shallower continental shelves result in a more pronounced upwelling surface
signal of denser water, albeit with reduced cross-shore transport and faster evolution compared to less inclined configurations
(Saldías and Allen, 2020).

In our experiments, the shallow shelf configuration led to a swift establishment of the downwelling front and increased
offshore transport, but showed greater susceptibility to instabilities compared to the deep shelf experiment. The deeper con-
figurations required more time for downwelling waters to form the front. Downwelling circulation resulted in a low Burger
Number, homogenizing conditions and reducing stratification. Unlike in upwelling scenarios, in downwelling conditions, the
shelf slope promoted downwelling flux, facilitating reduced stratification and vertical movements without significantly affect-
ing net vertical transport. Shallower configurations induced instabilities further offshore, while the canyon modified stability
characteristics, causing density variations. Upstream, minimal differences were observed, while downstream, density anoma-
lies varied, likely linked to changes in instability characteristics passing through the canyon.

The pattern of upwelling and downwelling around a submarine canyon is influenced by the angle between the canyon and the
current. Ahumada-Sempoal et al. (2015) observed a non-traditional pattern in Blanes Canyon, with offshore transport upstream
and onshore transport downstream, attributed to the angle between the canyon and the current. The interaction between the
current and the canyon is further influenced by their intersection, showing larger velocities when intersecting near the canyon
head compared to near the mouth (Alvarez et al., 1996; Jordi et al., 2005). In our simulations, driven solely by wind forcing,
the position of the downwelling front and related jet depends on wind magnitude and shelf slope. Responses were quicker
in the shallow shelf configuration and slower in the deep shelf scenario. However, the constant wind stress used may not
fully capture the realistic variability of winds over submarine canyons, which often experience oscillating downwelling and
upwelling-favorable winds in eastern boundary systems. Short-term fluctuations, influenced by factors like river effects on
density structures, can lead to varied downwelling responses in submarine canyons (Alvarez et al., 1996).

Differences in upwelling and downwelling responses between the upper shelf and within a canyon have been observed in
Mackenzie Canyon due to regional wind stress curl conditions (Lin et al., 2021). For instance, wind stress curl can induce up-
welling within the canyon even with locally downwelling-favorable winds, a dynamic not explored in this study. Furthermore,





previous conditions on the adjacent shelf can alter and amplify downwelling responses if they follow wind-driven upwelling
events, impacting the density structures advected into the canyon (Wang et al., 2022). Future experiments should consider var-
ious jet orientations, wind patterns, and density shelf conditions to comprehensively understand the impact of each component
on upwelling and downwelling dynamics around submarine canyons in more realistic settings.

Considering the impact of a submarine canyon on local circulation in a wind-driven downwelling regime, this local circula-
tion could play a crucial role in the biogeochemical conditions along the coast. While the overall contributions favor offshore
and downward movement, the local velocities influenced by the canyon could potentially reduce, trap, and even export particles
towards the coast, even under downwelling conditions. Our particle experiments indicated that a portion of particles can indeed
become trapped inside the canyon and be transported up to 8 km onshore. This phenomenon is more likely to occur with par-
ticles initially located inside the canyon or near its mouth, below the shelf-break depth. In such cases, particles may get caught
in an anticyclonic circulation and linger within the canyon for the 15 days of the tracking experiment. For particles carried by
the downwelling jet, some can be advected into the canyon and also entrapped by the anticyclonic circulation, a tendency that
is more pronounced in the shallow shelf experiment, which is the case with the largest percentage of particle trapped at the end
of the simulation. The trapping effect primarily arises from cross-shore velocities induced by the canyon, primarily observed
beneath shelf-break depth. Vertical velocities induced by the canyon do not lead to particle entrapment, primarily due to their
smaller magnitudes compared to the horizontal components.

While observations by Ahumada-Sempoal et al. (2015) around the head of the Blanes Canyon have shown similar trapping
mechanisms, the process is closely linked to the position of the jet relative to the canyon. This study suggests that our findings
offer a significant insight into particle trapping in canyons within Eastern Boundary Systems, regions where downwelling
conditions are predominantly wind-driven, rather than influenced by a consistent right-bounded current. Coupled with the
alternating favorable winds for upwelling and downwelling commonly observed in these systems, particles trapped during
storm events in canyons could potentially be further transported over the shelf by upwelling-favorable winds (Saldías and
Allen, 2020). Moreover, submarine canyons experiencing downwelling conditions have been associated with high productivity
(Skliris and Djenidi, 2006; Flexas et al., 2008). This productivity, along with the concentration of nutrients and non-mobile
organisms, has been linked to the preference of whales for using submarine canyons as habitats (Moors-Murphy, 2014). It
is essential for future observational studies to focus on how the predicted particle trapping in submarine canyons aligns with
actual observations to validate the real impacts of downwelling in the coastal ocean.

## 5  Conclusions

A series of process-oriented modeling experiments using three distinct bathymetric configurations driven by uniform downwelling-
favorable winds revealed that without the presence of a submarine canyon, isopycnals sank, forming a distinct downwelling
front in all three configurations, subsequently leading to the development of frontal instabilities. The introduction of a subma-
rine canyon altered the downwelling circulation, producing pronounced cross-shore velocities near the canyon with asymmetri-
cal patterns of onshore-offshore and downward-upward velocities on the upstream-downstream sides, respectively. The canyon



also influenced the propagation of baroclinic instabilities, affecting their wavelengths and structures downstream and resulting in density anomalies. The strong cross-shore velocities induced by the canyon extended to the surface, interacting with the Ekman surface layer and altering its vertical extension. While the overall impact of the canyon was to enhance offshore trans-
port through increased in cross-shore transport, there were also onshore velocities induced by the canyon on its upstream side. Additionally, an anticyclonic circulation inside the canyon could trap particles for the entire simulation period (15 days). This particle trapping phenomenon could affect both particles traveling along the continental slope with the incoming current and those initially located inside or over the canyon at the onset of the downwelling event. These results highlight the pivotal role of submarine canyons in counteracting the effects of downwelling regimes by opposing the offshore bottom Ekman transport
in the contiguous continental shelf. Future studies should further investigate the implications of this mechanism and consider more realistic wind forcing scenarios to enhance our understanding of the complex interactions between submarine canyons and downwelling conditions.

*Author contributions.*   GS and SA generated the idea to be study; PF undertook all the final analyses and wrote the original version of the manuscript. GS and SA reviewed and corrected the draft versión.

*Competing interests.*   The authors declare no competing interests.

*Acknowledgements.*   SA has been partially funded by NSERC Discovery RGPIN-2022-03112. We also thank to Compute Canada RAC 2022: Grant number RRG 1792 Digital Alliance of Canada RAC 2023: Grant number RRG 4603. GS was partially funded by Fondecyt 1220167 and COPAS Coastal ANID FB210021. GS also thanks the support from the National Laboratory for High Performance Computing (NLHPC).



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
