# Peer review of "The influence of a submarine canyon on the wind-driven downwelling circulation over the continental shelf"

_EGUsphere, 2024_

## Referee Comment (RC2)

General comments

This paper applies the ROMS ocean model to study the impact of downwelling favorable winds on canyons of different shelf bathymetries typical of Eastern Boundary systems and their influence on nearby flows and on offshore and downward transport. This study confirms and extends previous studies on downwellings in canyons using idealized modelling (in particular Spurgin and Allen, 2014). Interesting results are provided on the time-varying response of canyons to the forcing or on the ability of the canyons to trap a significant amount of particles under downwelling conditions. However, the results section is rather long and the novel results could be emphasized.

In terms of form, the paper is well written and clear. It includes extensive reference to the literature, which will be very useful to future research on the subject. I would therefore recommend this paper for publication if the following points are addressed.

Specific comments

**Methodology:**

l.74: Do you use sigma coordinates or generalized sigma-coordinates? The latests would be best suited to represent air-sea interactions, hence the wind forcing, and would allow for a better representation of the physical processes at the bottom of the canyon.

l.85: The bathymetries (and the following model description) are largely inspired by the work done by Saldias and Allen. Please add "Three types of bathymetric configurations**, already described by Saldias and Allen (2020)** were used".

l.122: please specify why you need to compute the topographic Burger number S.

**Results:**

l.149: "downward velocities occur upstream of the canyon". Please add on Figure 2d the upstream and downstream areas you are referring to. At the upstream corner, the velocity is upward. If your definition of upstream refers to y=-6 km to y=0 km as defined latter in the paper at l.253, one would find upward velocities on Figure 2d at the upstream canyon wall (red color) depending on the depth. This should be clarified as this statement is repeated several times in the manuscript.

l.165: Locating the downwelling front on the plot would help.

l.177: "Along the canyon axis (Y = 0 km), offshore and onshore velocities occur within the canyon". This seems obvious, do you mean velocity changes?

l.187: "values tend to diminish" and at l.189 "the numbers". Which values? Which numbers? These sentences may be reformulated.

l.269 and 272 : Please add "not shown" as the figure of dispersion of particles in the case NO-CANYON is not in the manuscript. I suggest this part to be shortened.

l.290: "the vertical velocities induced by the submarine canyon not significantly affecting the vertical particle movements". Can you explain this sentence please? I guess that particles follow

the water masses as they are passive tracers. At line l. 292 you mention an aggregation of particles. It should be mentioned if the particles behavior is taken into account with processes such as flocculation, else you should use another word like "accumulate".

l.296: What do you mean by "outside this range"?

**Discussion:**

l.318-322: The improvement in the wind forcing in this experiment compared with previous studies on downwelling canyons is emphasized in different parts of the manuscript, but it is not straightforward. Finally, what are the additional forcing terms in the equation? It is also not very clear to the reader what the novel results are in the study of downwelling canyons. What are the differences between the results of the present study compared to the previous ones?

**Conclusions:**

The conclusions part draws a clear picture of the downwelling canyon functioning, however it mixes original results emerging from this study with previous results that can be found in the literature. The novel results should be emphasized.

**Figures:**

Figure 2, Figure 6 nd Figure 7 (e,f): Please add the number of days after the start of the simulation when the plots were calculated.

Figure 3: "Cross-shore sections of velocity **field** (color) and **isopycnes** (gray lines) **at**". You should add the values of isopycnes on the plots or in the text.

Figure 4: "Cross-shore sections of velocity **field** (color) and **isopycnes** (gray lines) **at** 15 km (a-f) **downstream**, 0 km (g-l) **in the canyon** and -15 km (m-r) **upstream** for the no-canyon simulations at day 25.". You should add the values of isopycnes on the plots or in the text. For the plots at 0 km, the added canyon bathymetry should be represented using dotted lines and explained in the text.

Figure 5: Replace "density (gray lines)" with "isopycnes (gray lines)", and add the associated values. The location of the alongshore sections could be added on Figure 2 to help the reading of the paper. Note that in some of the plots in the manuscript, grey lines appear black rather than grey after printing, which is the case for this figure.

Figure 8: Upstream and downstream areas should be defined, either on Figure 2 as previously suggested, or here.

Figure 9: I guess that the red dotted lines are the isobaths, you should add it to the legend.

Typos

l.25: Something is missing in the sentence "for downwelling favorable flow (right/left-bounded in the Northern/South- ern Hemisphere) promote an anti-symmetrical circulation".

l.37: "focused **on** upwelling"

l.52: "is to **enhance** the downwelling" and "**These** biological characteristics"

Figure 2: "with (lower panels) **and** without"

l.137: "exten**d**"

l.147: "**at** the location"

l. 187: "In terms **of** stratification"

l.195:" Depending on **the** location"

l.215: "**at** x=-13"

l.219: "**due** to"?

Figure 7: A dot is missing at the end of the legend.

Figure 8: "(e-f) at z = -20 **m**"

l.271: There is a problem with this sentence, do you mean "Particles released in the presence of a canyon"?

l.281: "**Figure** 1"

l.296: "The percentage of **trapped** particles appeared to **increase**"

l.375: "**be trapped** by the anticyclonic circulation"

l.400: "increased cross-shore transport" or "an increase in the cross-shore transport"

l.408: "to be **studied**"

---

## Referee Comment (RC3)

General comments

This paper applies the ROMS ocean model to study the impact of downwelling favourable winds on the interaction of wind-driven current with a shelf-incising canyon for different configurations. After studying this work, I recommend revisions based on the following specific comments. In other journals this would be a resubmission with major revisions.

Specific comments

1. Missing literature

The literature review misses important previous work on downwelling including frontal instabilities. Here are some examples.

- Feliks and Ghil (1993) Downwelling-front instability and eddy formation in the Eastern Mediterranean. JPO, 23, 61-78.
- Brink, K,H. (2016) Continental Shelf Baroclinic Instability. Part I: Relaxation from Upwelling or Downwelling, JPO, 46, 551-568.
- Brink, K.H. (2024) The effect of alongshore wind stress on a buoyancy current's stability, CSR 272, 105149
- Kämpf, J. (2010) Extreme bed shear stress during coastal downwelling. Ocean Dynamics. https://doi.org/10.1007/s10236-019-01256-4

2. Missing discussion on timescales & type of model forcing

The wind-driven downwelling front develops over time and gradually moves away from the coast (see Kämpf, 2019). What are typical distances of upwelling fronts from the coast? How typical is it that a downwelling front actually comes close to a shelf-incised canyon? Given this, rather than a locally produced wind-driven upwelling front, wouldn't it make more sense to consider current-driven downwelling (driven by on offshore sea-level gradient) as forcing for your model? The results are probably different as this current could affect deeper portions of the canyon.

3. Frontal instabilities

3.1. Most results are affected by instabilities. These instabilities are key features of the results that require explanation, further analysis, and references to previous studies such as Feliks and Ghil (1993). But are these instabilities frontal instabilities? Some results of the cross-shelf velocity component u in Figure 3 and Figure 5 show negative values just above positive values near almost vertical isopycnals. These disturbances resemble overturning (i.e., forced convection) cells, rather than horizontal disturbances. Downwelling induces extreme situations of unstable density stratification. Could it be that the simulated instabilities are rather a side-effect of the vertical turbulence scheme used? The authors should explore whether the instabilities disappear when using difference turbulence schemes, grid resolutions and/or vertical density stratifications.

3.2. Another question: Why are the instability patterns horizontally tilted (see Figure 2)?

4. Methodology

Why is the model domain so big? The model domain shown in Figure 1b should have been more than sufficient for this investigation I would also recommend the use of cyclic boundaries as in

Brink's studies. A bottom roughness of 2 cm seems large. What happens for a bottom roughness of 2 mm?

5. Particle tracking

5.1. A key finding of this study is the trapping of particles inside the canyon. However, rather than distributing particles uniformly throughout the domain, particles were released along a few selected transects. Why were particles not distributed throughout the domain?

5.2. It seems the particle module was run standalone afterwards using the results of the ocean circulation model, true? Were the results stored after each simulation step (which requires massive amounts of storage), or was the output deemed stationary (which is not applicable given the instabilities and the progression of the downwelling front)? Or was the particle module run simultaneously with the ocean model? How many particles were released? I cannot find this information in the text.

5.3. While most particles are topographically steered across the canyon, the particles becoming trapped can hardly be seen in Figure 9. It would be better, in my view, to include a figure only displaying the trajectories of trapped particles. With arrows indicating flow directions. Other evidence should also be provided showing the anticyclonic eddy inside the canyon, e.g. as averaged horizonal current fields.

5.4. The percentage of trapped particles doesn't provide much useful information. The authors need to focus more on the reasons as to why the particles becomes trapped. Where do these particles exactly come from? At what depths were these particles released? Sometimes particles cam becomes trapped once there are too close to the seafloor (that's why diffusive effects sometimes help). Did this happen in the simulations? Does the inclusion of diffusive effects increase the trapping?

5.5. The discussion section refers to particles being released at mid-depths (e.g. 100-170 m). The use of the tern "mid depth" is incorrect and confusing. Instead of referring to the middle of water column, I think, the authors rather refer to a region where the total water depth ranges between 100 and 170 m. Please clarify this confusion.

---

## Author Comment (AC1)

**Response to reviewers' comments on the manuscript "The influence of a submarine canyon on the wind-driven downwelling circulation over the continental shelf: egusphere-2024-2386"**

Pedro A. Figueroa, Gonzalo S. Saldías and Susan E. Allen

October 2024

**1 Response to Reviewer #1**

**1.1 General Comments**

1. Finally, as the flow is strongly influenced by the presence of the submarine canyon, I think the authors should discuss the importance of an adequate resolution of the sigma layers at the bottom. This is an aspect of great relevance when performing realistic numerical simulations (see, for example, Clavel-Henry et al. 2019, Ocean Science, 15, 1745-1759)

Thank you for this point. We have clarified the sigma-coordinates parameters in the methodology, and we have added a paragraph in the discussion comparing to other papers focused on canyon/bottom boundary dynamics.

**1.2 Specific Comments**

1. L96-97: Since the numerical results depend on the correct definition of the bottom and surface Ekman layers, it is important to specify the scheme used to define the distribution of the sigma layers, indicating the values used for the relevant parameters, i.e. $\theta_a$, $\theta_b$, etc.

The information was added to the text as follows; "The experiments were configured with 30 s-coordinate levels, using an increased resolution near the surface and bottom to resolve the boundary layers, with $\theta_s = 3$ and $\theta_b = 1$, Vstretching = 4, Vtransform = 2 and hcline = 50 m."

2. L103-104: Include the wind direction for clarity.

Line 103 now indicates "uniform *northward (downwelling-favorable)* surface wind", indicating the meridional direction. Thank you.

---

## Author Comment (AC2)

**Response to reviewers' comments on the manuscript "The influence of a submarine canyon on the wind-driven downwelling circulation over the continental shelf: egusphere-2024-2386"**

Pedro A. Figueroa, Gonzalo S. Saldías and Susan E. Allen

October 2024

**1 Response to Reviewer #2**

**1.1 General Comments**

1. The results section is rather long and the novel results could be emphasized

    Thank you for your recommendation. The results section was shortened a little and some sentences were reformulated for clarity.

**1.2 Specific Comments**

1. l.74: Do you use sigma coordinates or generalized sigma-coordinates? The latests would be best suited to represent air-sea interactions, hence the wind forcing, and would allow for a better representation of the physical processes at the bottom of the canyon.

    Our simulations used generalized sigma coordinates following Shchepetkin & McWilliams(2009), which are the standard in ROMS models. Specifically, we used $\theta_s = 3$ and $\theta_b = 1$, Vstretching $= 4$, Vtransform $= 2$ and hcline $= 50$ m as input for the distribution of the vertical coordinates to ensure a good resolution in the upper and lower boundary layers. These parameters are now included in the revised version of of manuscript.

2. l.85: The bathymetries (and the following model description) are largely inspired by the work done by Saldias and Allen. Please add "Three types of bathymetric configurations, already described by Saldias and Allen (2020) were used".

    Thank you. We added this information to the text.

3. l.122: please specify why you need to compute the topographic Burger number S.

    The topographic Burger Number is computed to compare with previous and future canyon experiments. Even if it is not analyzed in detail in the text, is a common practice in canyon studies and help to compare with other studies including downwelling flows (e.g. Spurgin and Allen 2014).

4. l.149: "downward velocities occur upstream of the canyon". Please add on Figure 2d the upstream and downstream areas you are referring to. At the upstream corner, the velocity is upward. If your definition of upstream refers to y=-6 km to y=0 km as defined latter in the paper at l.253, one would find upward velocities on Figure 2d at the upstream canyon wall (red color) depending on the depth. This should be clarified as this statement is repeated several times in the manuscript.

A better description of the upstream areas has been done throughout the entire manuscript. Considering that the upstream wall of the canyon (which is where we are referring in l.149) refers to the coastal area between y = -6 km to y = 0, but the upstream area refers to any place south of the canyon axis, where the flow comes from (For example on the description of Figure 4, Y = -15 km is referred to as upstream). We have added the following statement to clarify this point:

"The presence of a submarine canyon induces notable changes in its vicinity (Fig. 2d,e,f), where upstream and downstream areas are defined as south and north of the canyon axis (Y = 0 km), respectively."

In addition, and for clarity, we added two boxes (in Figure 2d) denoting the Upstream and Downstream areas used in the calculations for Figure 9, using matching colors for clarity.

5. l.165: Locating the downwelling front on the plot would help.

Thank you for the comment. We have clarified the position of the front by indicating the labels of the specific isopycnals reaching the bottom. Labels of the isopycnals are included in the new version of those figures. This is now clarified in the text as well.

6. l.177: "Along the canyon axis (y = 0 km), offshore and onshore velocities occur within the canyon". This seems obvious, do you mean velocity changes?

Thank you for that comment. The sentence was not clear and the goal was to indicate that, compared to the no-canyon cases, velocities tend to erase the Ekman layer inside the canyon, which enhance the instabilities seen in the no-canyon cases. This has been rewritten as follow:

"Along the canyon axis (y = 0 km), offshore and onshore velocities within the canyon tend to overcome the typical flow of Ekman Layers (evident in the no-canyon cases in the deep shelf experiments). Moreover, instabilities extend into the canyon in the intermediate and shallow experiments, inducing strong cross-shore exchanges."

7. l.187: "values tend to diminish" and at l.189 "the numbers". Which values? Which numbers? These sentences may be reformulated.

This sentence was slightly changed considering that, as you mentioned, it was no clear what values we referred to. Now it clearly states the stratification and Burger number values:

"However, the stratification tends to diminish as the water column gets shallow over the continental shelf. This is evidenced from the Burger Numbers with $Bu = 0$ and $S = 0$ in the shallow shelf experiments..."

8. l.269 and 272 : Please add "not shown" as the figure of dispersion of particles in the case NOCANYON is not in the manuscript. I suggest this part to be shortened.

"*Not shown*" has been added to the corresponding lines describing the no-canyon cases. The description was slightly shortened considering your general comment. Thank you.

9. l.290: "the vertical velocities induced by the submarine canyon not significantly affecting the vertical particle movements". Can you explain this sentence please? I guess that particles follow the water masses as they are passive tracers. At line l. 292 you mention an aggregation of particles. It should be mentioned if the particles behavior is taken into account with processes such as flocculation, else you should use another word like "accumulate".

Here we refer to the vertical velocities shown in Figure 2, and how the velocities, which are a direct effect of the canyon on the circulation, do not affect significantly to the vertical movement of particles in the experiments. The word aggregation was changed to accumulate, since these particles are just neutrally buoyant without any extra behavior, as you mentioned. Very good point to clarify. Thank you.

10. l.296: What do you mean by "outside this range"?

"Outside this range" was referring to particles that were released above or below of the depth ranges indicated in Figure 9. This has been rewritten for clarity:

"Outside of the depth ranges showed in Figure 9 (i.e. up to the surface and down to 400 m depth)"

11. l.318-322: The improvement in the wind forcing in this experiment compared with previous studies on downwelling canyons is emphasized in different parts of the manuscript, but it is not straightforward. Finally, what are the additional forcing terms in the equation? It is also not very clear to the reader what the novel results are in the study of downwelling canyons. What are the differences between the results of the present study compared to the previous ones?

The simulations of this work does not add extra forcing terms to the primitive equations solved by ROMS. Moreover, the simulations are driven solely putting a wind stress on the surface of the model, without body forces, as it has been typically done. The emphasis is to reproduce wind-driven downwelling conditions (including a downwelling front and jet, etc) which are typical for mid-latitude eastern boundary margins during winter. This is why we also include no-canyon experiments which do resemble the results of wind-driven downwelling circulations from previous studies (e.g. Austin and Lentz, 2002).

12. The conclusions part draws a clear picture of the downwelling canyon functioning, however it mixes original results emerging from this study with previous results that can be found in the literature. The novel results should be emphasized.

The conclusions' section was rewritten and separated in two parts. The first one addresses the results that we reproduce and are consistent with previous studies. The second part highlights the novel results related to the modification of instabilities by the canyon, the vertical extension of the cross-shore velocities induced by the canyon and how they modify surface and bottom Ekman layers. Emphasis is given in the particle trapping as a result of the lagrangian trajectories of the flow. Thank you.

13. Figure 2, Figure 6 and Figure 7 (e,f): Please add the number of days after the start of the simulation when the plots were calculated.

Indication of the days were added to the description of Figures 2 and 6. In the case of Figure 7(e,f), the patterns shown correspond to the integrated transport since day 10 up to day 25, which is now described as well.

14. Figure 3: "Cross-shore sections of velocity field (color) and isopycnes (gray lines) at". You should add the values of isopycnes on the plots or in the text.

   Isopycnal values were added to Figures 3-4. Thank you.

15. Figure 4: "Cross-shore sections of velocity field (color) and isopycnes (gray lines) at 15 km (a-f) downstream, 0 km (g-l) in the canyon and -15 km (m-r) upstream for the no-canyon simulations at day 25.". You should add the values of isopycnes on the plots or in the text. For the plots at 0 km, the added canyon bathymetry should be represented using dotted lines and explained in the text.

   Isopycnal values were added also to Figure 4 and the bathymetry of the canyon is now showed on black dashed lines. This is described in the legend of the figure too.

16. Figure 5: Replace "density (gray lines)" with "isopycnes (gray lines)", and add the associated values. The location of the alongshore sections could be added on Figure 2 to help the reading of the paper. Note that in some of the plots in the manuscript, grey lines appear black rather than grey after printing, which is the case for this figure.

17. Density was changed for isopycnal and values were added to the figure. In Figure 2, green lines were added to show the location of the alongshore sections of Figure 5 and Figure 6.

18. Figure 8: Upstream and downstream areas should be defined, either on Figure 2 as previously suggested, or here.

19. Areas were defined in Figure 2 following previous comments.

20. Figure 9: I guess that the red dotted lines are the isobaths, you should add it to the legend.

   A description has been added to the legend of the figure "Isobaths of the submarine canyon (from 100 to 400 m) are shown in dashed red lines". Thank you for this comment.

21. Typos:

   - l.25: Something is missing in the sentence "for downwelling favorable flow (right/left-boundedin the Northern/South- ern Hemisphere) promote an anti-symmetrical circulation".

     It has been written again as "In contrast, downwelling favorable flow (right/left-bounded in the Northern/Southern Hemisphere) promotes an anti-symmetrical circulation".

   - l.37: "focused on upwelling"

     It has been corrected.

   - l.52: "is to enhance the downwelling" and "These biological characteristics"

     It has been corrected.

   - Figure 2: "with (lower panels) and without"

     It was added to the figure description.

   - l.137: "extend"

     It has been done.

   - l.147: "at the location"

     It is corrected now.

- l. 187: "In terms of stratification"

  This sentence was rewritten.

- l.195:" Depending on the location"

  Yes. It is corrected now.

- l.215: "at x=-13"

  changed "in" for "at"

- l.219: "due to"?

  Yes. Corrected now.

- Figure 7: A dot is missing at the end of the legend.

  Thank you, it was included in this revised version.

- Figure 8: "(e-f) at z = -20 m"

  It has been corrected.

- l.271: There is a problem with this sentence, do you mean "Particles released in the presence of a canyon"?

  It has been rewritten as "Particles in simulations with a canyon".

- l.281: "Figure 1"

  Thank you for this one. It was added "Figure 1" for consistency

- l.296: "The percentage of trapped particles appeared to increase"

  It has been corrected.

- l.375: "be trapped by the anticyclonic circulation"

  It has been corrected.

- l.400: "increased cross-shore transport" or "an increase in the cross-shore transport".

  Please check the sentence was rewritten as "Moreover, while the net effect of the canyon is to enhance downwelling, local onshore velocities in the vicinity of the canyon lead to an anticyclonic circulation inside the canyon that can trap particles for the entire simulation period (15 days)".

- l.408: "to be studied"

  Not really sure to which line you referred to (the last line was the 407). Nonetheless, we have re-written the last section of the conclusions to make sure it is clear for the reader.

---

## Author Comment (AC3)

**Response to reviewers' comments on the manuscript "The influence of a submarine canyon on the wind-driven downwelling circulation over the continental shelf: egusphere-2024-2386"**

Pedro A. Figueroa, Gonzalo S. Saldías and Susan E. Allen

October 2024

**1 Response to Reviewer #3**

**1.1 Specific Comments**

1. The literature review misses important previous work on downwelling including frontal instabilities. Here are some examples.

   Thank you. Your recommendations were added to the introduction section. Other relevant references regarding wind-driven downwelling processes have been also included.

2. The wind-driven downwelling front develops over time and gradually moves away from the coast (see Kämpf, 2019). What are typical distances of upwelling fronts from the coast? How typical is it that a downwelling front actually comes close to a shelf-incised canyon? Given this, rather than a locally produced wind-driven upwelling front, wouldn't it make more sense to consider current-driven downwelling (driven by on offshore sea-level gradient) as forcing for your model? The results are probably different as this current could affect deeper portions of the canyon.

   We appreciate your comment. Nonetheless, the focus of the paper is to address the response of a downwelling front, and the associated circulation, generated in response to the wind forcing. This be an important part of the dynamics of mid-latitude Eastern Boundary systems, where local storms events could induce strong events of downwelling (Austin & Barth 2002). These storm can tilt isopycnal from the coast offshore up to about 30-40 km. Please note that we have reproduced the typical wind-driven downwelling results from other studies in our no-canyon experiments. Thus, introducing the submarine canyon in our simulations allowed us to highlight the impact of having a submarine canyon in the coastal circulation of an idealized eastern boundary margin during winter conditions.

   We have included some literature of the case you mention including a current-driven downwelling in realistic (Jordi et al. 2005) and idealized cases (Klinck 1996, Spurgin & Allen 2014). In this revised version we have also included new references of your studies (Kämpf, 2006, 2007, 2009, 2012, 2018, 2019). Thank you.

3. . Most results are affected by instabilities. These instabilities are key features of the results that require explanation, further analysis, and references to previous studies such as Feliks and Ghil

(1993). But are these instabilities frontal instabilities? Some results of the cross-shelf velocity component u in Figure 3 and Figure 5 show negative values just above positive values near almost vertical isopycnals. These disturbances resemble overturning (i.e., forced convection) cells, rather than horizontal disturbances. Downwelling induces extreme situations of unstable density stratification. Could it be that the simulated instabilities are rather a side effect of the vertical turbulence scheme used? The authors should explore whether the instabilities disappear when using difference turbulence schemes, grid resolutions and/or vertical density stratifications.

We appreciate your comment. The instabilities appear just at the location of the downwelling front. We have attached a complementary animation showing the evolution of the cross-shore and vertical velocity in the x-z plane. As it can be seen there, isopycnal in the shallow shelf case (which is the one that most rapidly evolves) does not present isopycnals going into unstable distributions. Rather, they form a front near the bottom and then start to oscillate as baroclinic instabilities develop, with vertical velocities in the entire column. We believe these are not representing overturning. The downwelling wind forcing is not that strong and the lighter waters does not seem to penetrate below the denser water. Thank you for pointing out to the paper by Feliks and Ghil (1993) – great reference to include in the introduction and discussion.

You can download the movie at Downwelling Movie Isopycnal. It is also reproduced here:

4. Why are the instability patterns horizontally tilted (see Figure 2)?

Considering that the northward flow is horizontally and vertically sheared around the core of instabilities in both, the intermediate and shallow shelf experiments (Figure 2 and Figure 3), it is expected that the perturbations related to the frontal instabilities should be advected not uniformly, thus generated the tilted pattern observed in Figure 2. Tilted patterns of frontal instabilities have been observed for a similar bathymetric setup in Durski & Allen (2005) and in relation to a vertically sheared alongshore flow.

5. Why is the model domain so big? The model domain shown in Figure 1b should have been more than sufficient for this investigation I would also recommend the use of cyclic boundaries as in Brink's studies. A bottom roughness of 2 cm seems large. What happens for a bottom roughness of 2 mm?

The model domain is the same as Saldías & Allen (2020), and thus, their explanation applies here as well. Considering that the objective is to generate a wind-driven circulation with impose perturbations on the forcing to generate instabilities, the use of periodic boundaries could allow the propagation of waves and noise through the domain in a constant loop. Previous works have also noticed that in domains with submarine canyons, the propagation and reflection of waves make open boundaries a better choice versus periodic boundaries (e.g. Dinniman & Klinck, 2002; Zhang & Yankovsky, 2016). The study of Saldías et al. (2021) shows that the interaction of Coastal Trapped Waves with a canyon can promote the generation of new modes which also continue propagating along the coast. Thus, we opt for the configuration having a large domain with open boundaries, similar to other studies (e.g. Zhang & Lentz, 2017). The bottom roughness was computed following previous studies of flow over topography using ROMS (e.g. Whitney & Allen, 2009; Li et al., 2021).

6. A key finding of this study is the trapping of particles inside the canyon. However, rather than distributing particles uniformly throughout the domain, particles were released along a few selected transects. Why were particles not distributed throughout the domain?

Thank you for the comment. The goal of the particle tracking experiment is to give some extra information on the circulation patterns inside the canyon from a Lagrangian perspective. With this in mind, we selected upstream positions to evaluate if particles outside of the canyon can enter and become trapped. The transects over the canyon look for particles that start in the canyon and remains in there. Overall, this approach of discrete transects allows us to show that particles near the canyon can become trapped, and use this in the future for further development. It also allows for a better observation compared to releasing particles along the whole domain.

7. It seems the particle module was run standalone afterwards using the results of the ocean circulation model, true? Were the results stored after each simulation step (which requires massive amounts of storage), or was the output deemed stationary (which is not applicable given the instabilities and the progression of the downwelling front)? Or was the particle module run simultaneously with the ocean model? How many particles were released? I cannot find this information in the text.

The particles were run standalone using the velocities of the model as input for the lagrangian trajectories. For each time, the velocities of the model were interpolated on z-level coordinates with a resolution of 2.5 meters, and then this interpolated fields were used as input for Parcels (Delandmeter et al., 2019). The particles were released from surface to bottom at the indicated transects (locations indicated with white dots in Figure 9). These details of particle trajectories and the use of Parcels is included in the revised version of the manuscript. Thank you.

8. While most particles are topographically steered across the canyon, the particles becoming trapped can hardly be seen in Figure 9. It would be better, in my view, to include a figure only displaying the trajectories of trapped particles. With arrows indicating flow directions. Other evidence should also be provided showing the anticyclonic eddy inside the canyon, e.g. as averaged horizontal current fields.

We have added a new figure highlighting the initial position of the particles trapped in the canyon at the end of the simulation. Thank you for pointing this out. We believe this new figure clarifies this trapping effect.

9. The percentage of trapped particles doesn't provide much useful information. The authors need to focus more on the reasons as to why the particles becomes trapped. Where do these particles exactly come from? At what depths were these particles released? Sometimes particles cam becomes trapped once there are too close to the seafloor (that's why diffusive effects sometimes help). Did this happen in the simulations? Does the inclusion of diffusive effects increase the trapping?

As mentioned before, the panels of Figure 9 show the initial position of the particles (white points). The depth ranges indicated in the title of each panels indicate the range of release of the particles. In our case, any particle that become trapped in the seafloor or in the walls of the canyon were discarded from the analysis. We did not analyze the processes that trap the particles in detail, as it would extent the paper way beyond the scope of the manuscript. Nonetheless, it is a good point to continue with further experiments in a future study.

We included new information in the manuscript regarding the particle tracking experiments:

"Velocity fields from ROMS were interpolated at each time step to z-levels from 5 m to 400 m with a 2.5 m resolution. These velocity fields were used as input to the Parcels environment. Particles that were trapped by the seafloor or stopped by the topography were discarded from our analyses, and thus, we only maintain particles that presented displacements through the entire period of tracking."

10. The discussion section refers to particles being released at mid-depths (e.g. 100-170 m). The use of the tern "mid depth" is incorrect and confusing. Instead of referring to the middle of water column, I think, the authors rather refer to a region where the total water depth ranges between 100 and 170 m. Please clarify this confusion.

To clarify any potential confusion, the term Mid-depth was changed along the text for the corresponding depth interval, or a more general conditions such as "depths below the continental shelf". We want to thank the reviewer, Dr. Jochen Kämpf, for his comments.

Please note that all the references we have specified here are included in the revised version of the manuscript.